# The Effect of Space Objects on Ionospheric Observations: Perspective of SYISR

**Junyi Wang** [1,2,3] , **Xinan Yue** [1,2,3,*] , **Feng Ding** [1,2,3] , **Baiqi Ning** [1,3] , **Lin Jin** [4] , **Changhai Ke** [4] , **Ning Zhang** [1,3] , **Junhao Luo** [1,3] , **Yonghui Wang** [1,2,3] , **Hanlin Yin** [1,2,3] , **Mingyuan Li** [1,2,3] and **Yihui Cai** [1,2,3]

1   Key Laboratory of Earth and Planetary Physics, Institute of Geology and Geophysics, Chinese Academy of Sciences, Beijing 100029, China
2   College of Earth and Planetary Sciences, University of Chinese Academy of Sciences, Beijing 100049, China
3   Beijing National Observatory of Space Environment, Institute of Geology and Geophysics, Chinese Academy of Sciences, Beijing 100029, China
4   Nanjing Research Institute of Electronics Technology, Nanjing 210039, China
*   Correspondence: yuexinan@mail.iggcas.ac.cn

**Abstract:** Space objects around the Earth are a potential pollution source for ground-based radio observations. The Sanya incoherent scatter radar (SYISR) is a newly built active digital phased array, all solid-state transmitting and digital receiving incoherent scatter radar in Sanya (18.3°N, 109.6°E), with the main purpose of ionospheric monitoring and investigations. In this study, we presented the effect of the greatly increased number of space objects on ionospheric observations through SYISR. Firstly, we showed the space object pollution on the range-time-intensity (RTI), autocorrelation function (ACF)/power spectra, and ionosphere parameter of SYISR measurements. An altitude of around 600 km is the region where space objects occur most frequently. Then, we eliminated the space object pollution using the traditional smallest of constant-false-alarm-rate (SO-CFAR) algorithm. However, pollution from smaller space objects remains, whose reflected echo is comparable to or lower than the background ionosphere, which results in unrealistic retrieved ionospheric electron density. Furthermore, we quantitatively assessed the space object effect based on the current space object orbit database and simulation. The pollution should linearly increase with the increase in the number of space objects in the future. Among the space objects, whose radar cross section (RCS) and orbit information are now published, there still exist ~9000 (~37% of the total number) space objects, whose effect is difficult to eliminate. This study is beneficial to the data process of SYISR and has implications for similar types of ionospheric observations by radar.

**Keywords:** incoherent scatter radar; space object pollution; ionospheric observation

## 1. Introduction

Space objects around the Earth encompass artificial satellites, rocket bodies, human-made orbital debris, and natural meteoroids. Orbital debris mainly comes from artificial satellites, which directly increase the number of space objects in orbit. The satellite constellation collects the satellites that share a common design, distributed among different orbits to provide a service and/or land coverage that cannot be achieved by only one single satellite [1]. Their applications range from telecommunications, meteorology, and remote sensing to global navigation satellite systems. In the early space age, a constellation usually had dozens of satellites. In recent years, a series of large constellation plans has been approved around the world, such as "StarLink", "Kuiper", "OnWeb", and "StarNet" [2]. These constellations propose to consist of tens of thousands of artificial satellites. If they are running on schedule, there will be an explosively increasing trend of space objects in low Earth orbit (LEO) in the next decade. The dense space objects around the Earth represent potential pollution by interfering with astronomical ground-based observations of both the optical and the radio kind [3,4]. Scientists have assessed the brightness level of orbiting

bodies in optical observations based on the StarLink constellation [5,6]. Meanwhile, simulations have been performed to analyze satellite light pollution with respect to geographic location and time [7–9]. Similar problems have been faced in radio observations. The disturbances from the space objects in ISR's ionosphere measurements are well-known [10,11]. An automated detection method of satellite contamination has been introduced to check the data for anomalous spectra and isolate those cases [11]. Unlike scattering sunlight in optical observations, the space objects receive and scatter the transmitted radio signal when they are within the radar beam. If the scattering signal is sufficiently large to be detected, the space objects will appear in the original signal and causes an abnormal radar echo. If the pollution caused by space objects too frequently occurs, it will cause the failure of observations.

Incoherent scatter radar (ISR), usually with high power and antenna gain, plays an important role in ground-based radio facilities for ionosphere research. High-performance incoherent scatter radars usually also have excellent space object detection capabilities and can find centimeter-level space objects at a kilometer height without integration [12–14]. The Sanya incoherent scatter radar (SYISR, 18.3°N, 109.6°E) is the new generation of incoherent scatter radar developed by China [15,16]. It works at the ultra-high frequency band (~440 MHz). The active digital phased array antenna of SYISR realizes a microsecond-level beam switching capability by digital beam forming technology, which making the radar capable of fast wide-region detection. To ensure high-quality ionosphere results, the space object pollution in the radar measurements cannot be ignored. It is necessary to identify the impact of space object pollution on SYISR ionosphere observations and handle the obvious effects. Meanwhile, the assessment of the effects from large constellation satellites is important for understanding and handling the potential threats of this type of pollution, which is the main aim of the paper.

This section summarizes the motivations of our study. In Section 2, we describe in detail the space objects found in the ionosphere observations when we used different waveforms during the experiment. To eliminate the effect of space object pollution, we adopt a special process to remove the obvious space objects from the data, which is shown in Section 3. Furthermore, we analyze the impacts of the increasing number of satellites, due to the StarLink constellation, on SYISR observations. The size of space objects with the risk of pollution after the special removal process is presented in Section 4.

## 2. Space Objects in the SYISR Ionospheric Observations

The ionosphere observation of ISR is based on Thomson scattering from the randomly distributed free electrons in the ionosphere [17,18]. Although the scattered signal power from a single electron is low, scientists have predicted and confirmed that a powerful radar with high gain and a large aperture can measure the ionosphere by integrating the signal from a scattering volume at a certain time period [19,20]. When the space objects pass through the ISR radar beam, the abnormal echo will pollute the power profile and autocorrelation function (ACF)/power spectrum of the measured signal and consequently pollute the inverted ionosphere parameters (including the electron density, electron temperature, ion temperature, and line-of-sight velocity).

In the ISR signal process, incoherent integration is usually employed to improve the signal-to-noise ratio (SNR) and the statistical accuracy of the estimation. There is some flexibility with concern to the integration time because SYISR saves the original I/Q data. A longer integration helps to obtain a better statistical accuracy of the measurements under the assumption of a stable ionosphere. If the temporal resolution is smaller than the time that the space object passes through the radar beam, the space object will appear in several consecutive time intervals. For the space objects among the ionosphere observations, the echo power that we receive is related to the size of these space objects and the range in which they are located. Figure 1 shows an overview of the appearance of space objects in the RTI plot of SYISR ionospheric observations in three experiments with different typical waveforms. The long pulse (LP) and alternating code (AC) signal experiments occurred

simultaneously because we used combined coding to execute both in one experiment. To demonstrate the effect of integration time, we have shown the results with 3.2-s and 32-s of integration for comparison in Figure 1. There are ~20 (10) obvious space objects identified with the 3.2-s (32-s) integration time in both experiments. Generally, the SNR is ~10–30 dB higher around the space object occurrence location, and obvious pollution can be identified. The ionospheric echo power is higher with the 32-s integration than with the 3.2-s integration because echoes from more electrons are added. For the SNR around the space object occurrence, the value does not change with the integration time. With 32-s of integration, some smaller space objects become nonvisible because their SNRs are comparable with the background ionosphere after a longer integration time, especially around the ionospheric peak altitude. The range extension of the space object is related to the pulse width, since we integrated before decoding, which is ~72 km for the LP/AC results and ~58.5 km for the barker code (BK) results in Figure 1. The temporal extension of the space object in Figure 1 is related to the temporal resolution, which of the LP/AC combination experiment is ~2 times larger than that of the BK experiment.

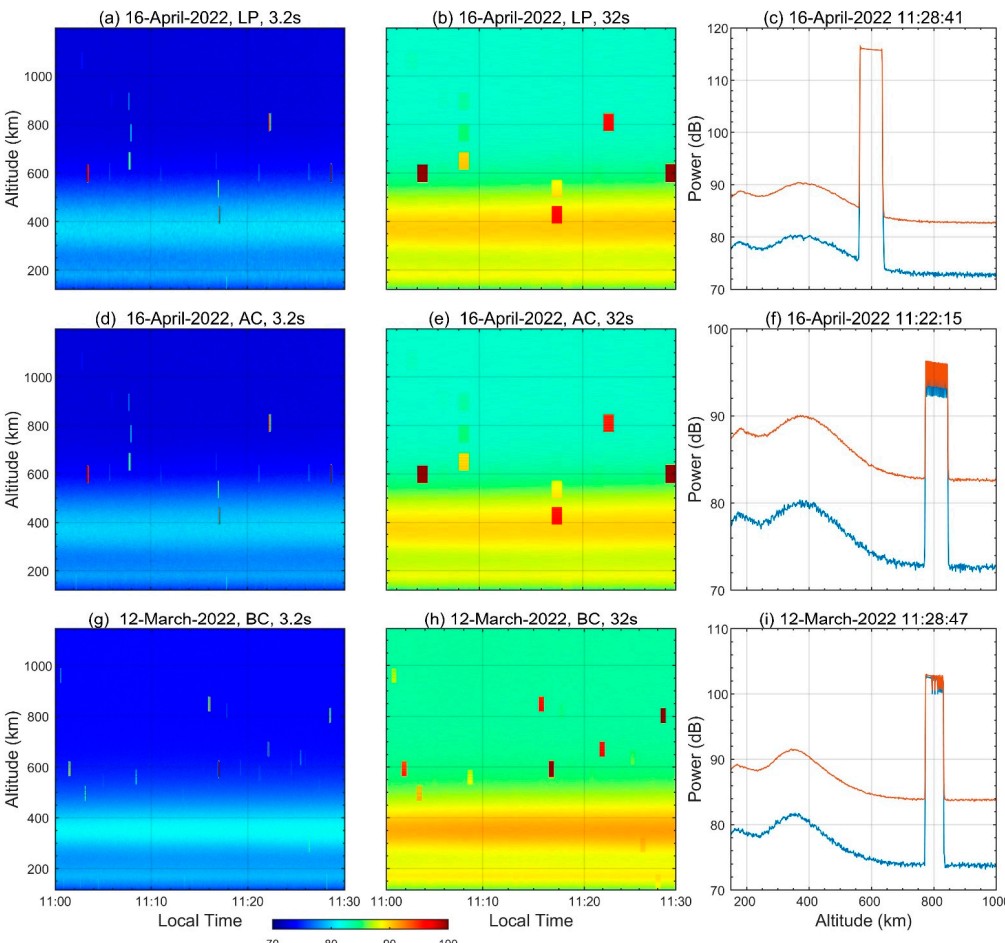

**Figure 1.** Range-time-intensity (RTI, unit: dB) of the SYISR observations with different integration times (3.2 s in the left column, 32 s in the middle column) during half-hour intervals of 16 April 2022 (upper and middle panels) and 12 March 2022 (lower panel). The right column is the corresponding power profile case with a space object pollution during a selected time (blue line: results with the 3.2-s integration, red line: results with the 32-s integration). The upper (**a**–**c**), middle (**d**–**f**), and bottom (**g**–**i**) panels are the long pulse (LP, 480 μs), 16-bit alternating code (AC, 480 μs) signal, and 13-bit barker code (BK, 390 μs) signal, respectively. The radar beam steered to the zenith direction ranging from 120 to 1200 km in all three experiments.

During the ionospheric parameter retrieval from the ISR signal, the ACF is calculated after decoding [21–23]. The corresponding power spectra can be derived from the ACF through the Fourier transform. If the space object pollution is not eliminated properly in the original data, it will affect the ACF and power spectra calculation. As an example, Figure 2 shows the ACF and power spectra profiles derived from SYISR measurements on 16 April 2022 at 11:23:15. In this case, the space objects appear at altitudes of approximately 381 km and 488 km. As indicated, the space object occurrence results in a strong amplitude in both ACF and power spectra at the corresponding altitude and makes the results unreliable and unusable. Its range extension is approximately 72 km in the LP signal with a pulse length of 480 μs. We use the 16-bit AC signal with the same pulse length as the LP signal. The range extension after decoding reduces to approximately 4.5 km, which is equal to one baud length. The range resolution using this AC signal is 16 times shorter than the LP.

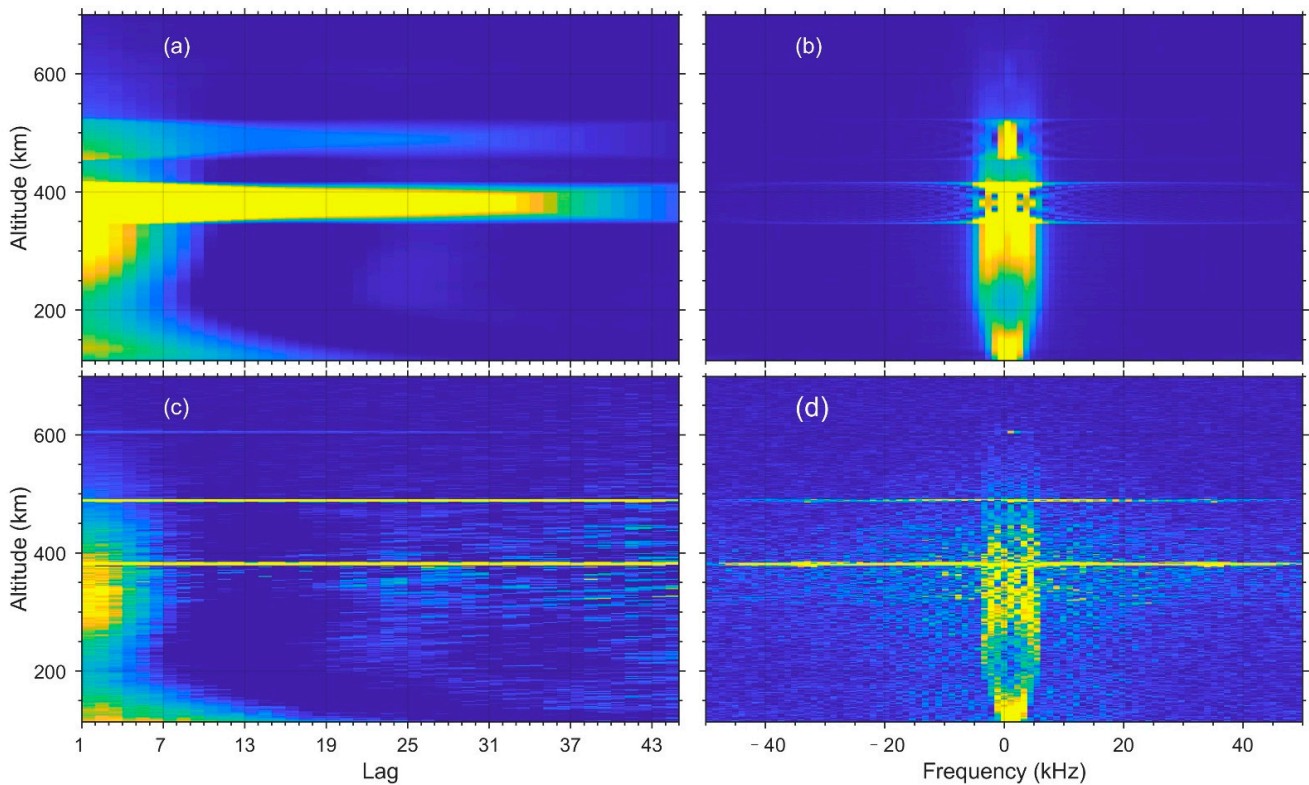

**Figure 2.** Autocorrelation function (ACF, (**a**,**c**)) and the corresponding power spectrum (**b**,**d**) of SYISR ionospheric observation with obvious space object pollution on 16 April 2022 at 11:23:15. The top panel shows the results using the long pulse (LP) signal, while the bottom panel shows the results using the alternating code (AC) signal.

## 3. Preliminary Attempts to Eliminate Space Object Pollution

A large amount of space object pollution reduces the quality of the ionosphere results. A special process to handle the effect of space objects on observations is necessary. If the echo is lower than the detection ability of radar, the corresponding space objects cannot be found in the observations, and this type of pollution is neglected. In this study, we implemented the commonly used smallest of constant-false-alarm-rate (SO-CFAR) detector algorithm to identify and eliminate the space objects in our data process [24–27]. Specifically, we found the abnormal echoes of space objects at different ranges. The detection of the space objects was completed in the step when we obtained the power profiles of the pre-integration pulses, which is a part of the SYISR data process to obtain the ionosphere parameters. If the space objects were detected in a certain range, the raw IQ data at the according range would be replaced. The neighboring information of the range is used to

estimate the detection background. For space object detection in ionosphere observations, the number of reference units used is important. Instead of the usual space object detection in a stable background, the background includes ionosphere variability. The fluctuation of the ionosphere affects the background estimation in different reference unit numbers. From the results in Figure 1, the ionosphere has a different spatial distribution from space objects. Ionospheric echoes appear over a wide and continuous range, while the appearance of objects in space is usually within a range resolution. Therefore, the width of the reference units should not be too wide. In our process, we chose a detection reference slightly wider than one pulse to simultaneously avoid the multi-target ambiguity and the misjudgment of the ionosphere. However, at the beginning and end scope, these reference units are not sufficiently wide because of the lack of the upper or lower side, which causes missed detection at the start and the end of the signal. We increased the reference units by two times in this situation.

Another important factor that affects the correctness of space object detection in the ionosphere observations of SYISR is the detection threshold. The detection threshold is related to the false alarm probability. The false alarm probability means the rate at which the false detection of space objects occurred. Different detection thresholds and reference units were tested to find the balance between protecting the ionosphere echo and picking out the space objects. A higher unsuitable threshold will increase the possibility of missed detections, especially when the space objects occur at the altitude of the F layer. Because the space objects are mixed with the ionosphere echo, the difference between the power of space objects and the background of the ionosphere is small. With a high detection threshold, the missed detection of space objects is probable. A lower unsuitable threshold will increase the number of false alarms. In a false alarm, an ionosphere echo with strong fluctuation is regarded as space object pollution and incorrectly removed. To avoid this situation, we consider the fluctuation level of the ionosphere to obtain a suitable threshold. The background is divided into high background and low background according to the altitude of 600 km. In the low background, there is mainly environmental noise, which hardly changes with the range, and its power is relatively stable and small. We set the detection threshold as a lower value to reduce missed detections. In the high background, the ionosphere echo is relatively stronger than the normal environmental noise. The detection threshold increases to two times that in the low background. According to the detection threshold and the detection reference unit number used, the theoretical false alarm probability was about 2% in the high background and 10% in the low background. Furthermore, because the number of reference units was set to change with the sampling points of one pulse width, there were some differences when using different waveforms. As for the measurements from different azimuths and elevations, the range was changed to the altitude for the separation of the backgrounds. The method was applied in the same way as when the radar was pointing to the zenith. We performed a space object elimination test on both the LP (11 March 2022) and AC (14 April 2022) signals from the entire-day SYISR observations. Figure 3 shows the RTI before and after elimination, and their difference during the AC experiment. The error of the raw electron density before and after the removal of space objects is also presented. Obvious space objects are successfully deleted from the ionosphere observations while reserving the normal ionospheric echo. The neighboring data at the same range of different pulses in the time domain are used to fill in the lost signal polluted by the space objects. The data used to fill in the missing signal were measured within the pre-integrated time (usually a few seconds). Considering that the incoherent integration of the ionosphere echo was applied under the assumption of a stable ionosphere during the integration time, we regarded the data from the neighboring pulse within this time as the repeated measurements for the ionosphere. Then the data were used as a substitute for the ionosphere echo polluted by the space objects. To demonstrate the effect of the SO-CFAR algorithm for space object elimination on the subsequent data inversion, we compare one specific power profile and the ACF, and the retrieved electron density profile during the entire day of 14 April 2022 before and

after space object elimination (Figure 4). As shown in the ACF results, the decoding of the AC signal improves the range resolution to a narrower width (about one baud length) in the ionosphere measurements. The disturbances of space objects were still found over a range extension after decoding. The reason for this was that the power of the space objects would change with time. The range extension of space objects was left over even after the decoding process was finished. When the least squares fitting was applied to obtain the ionosphere parameters from the measured spectrum (FFT from the ACF), space object pollution was present over a range extension in the fitted ionosphere parameters. When the detected space objects were removed from the raw IQ data, we can see that the effect of the space object was successfully eliminated in the power profile, the ACF profile, and the electron density profile. In total, 389 space objects were successfully identified and removed during the entire day.

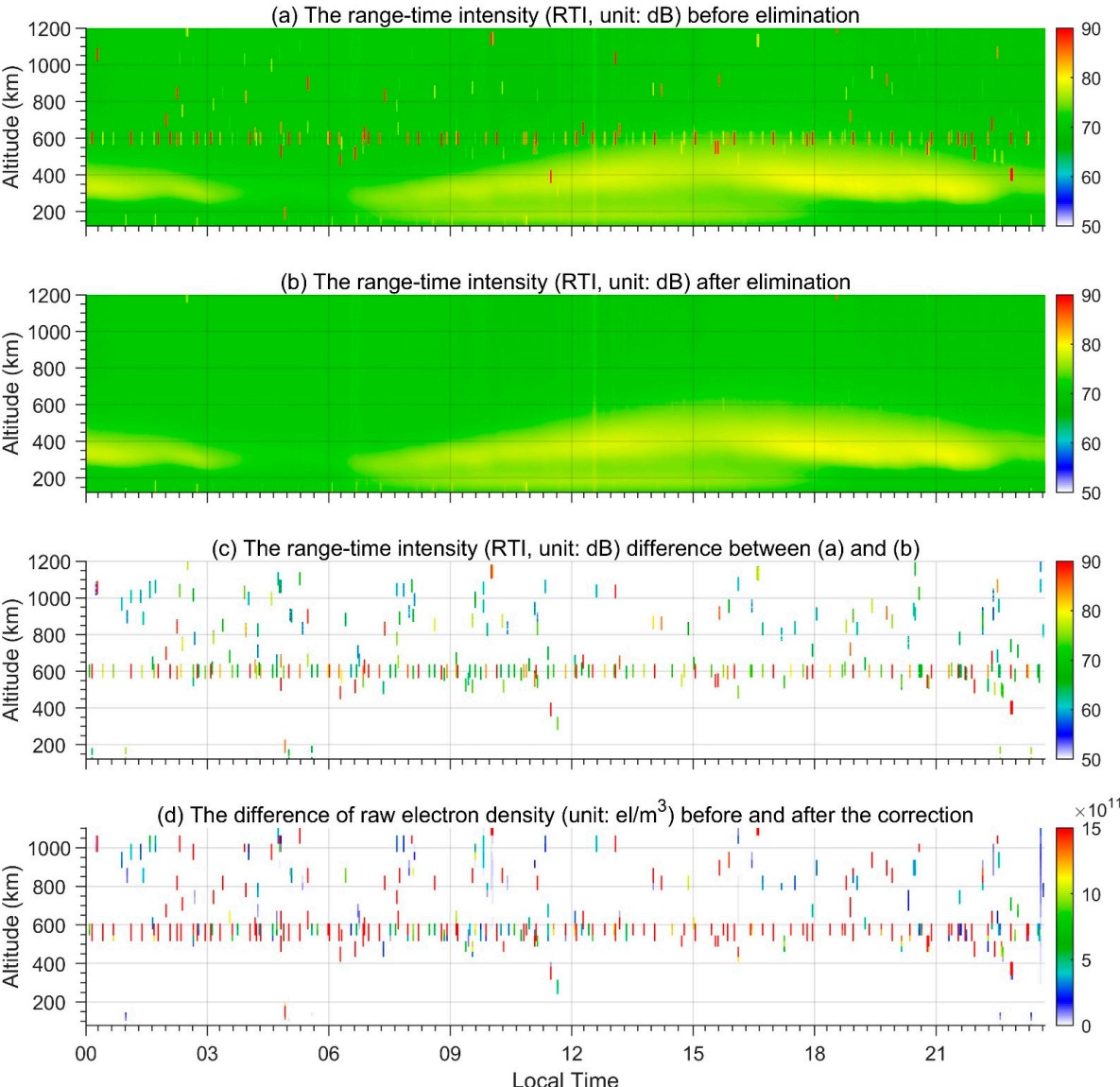

**Figure 3.** Range-time-intensity (RTI, unit: dB) of the ionosphere observations during 14 April 2022 using the alternating code (AC) signal before (**a**) and after (**b**) eliminating the space object pollution using the SO-CFAR algorithm. Panel (**c**) is the difference between (**a**) and (**b**), which indicates the identified space object occurrences. Panel (**d**) is the error of the raw electron density before and after the removal of space objects.

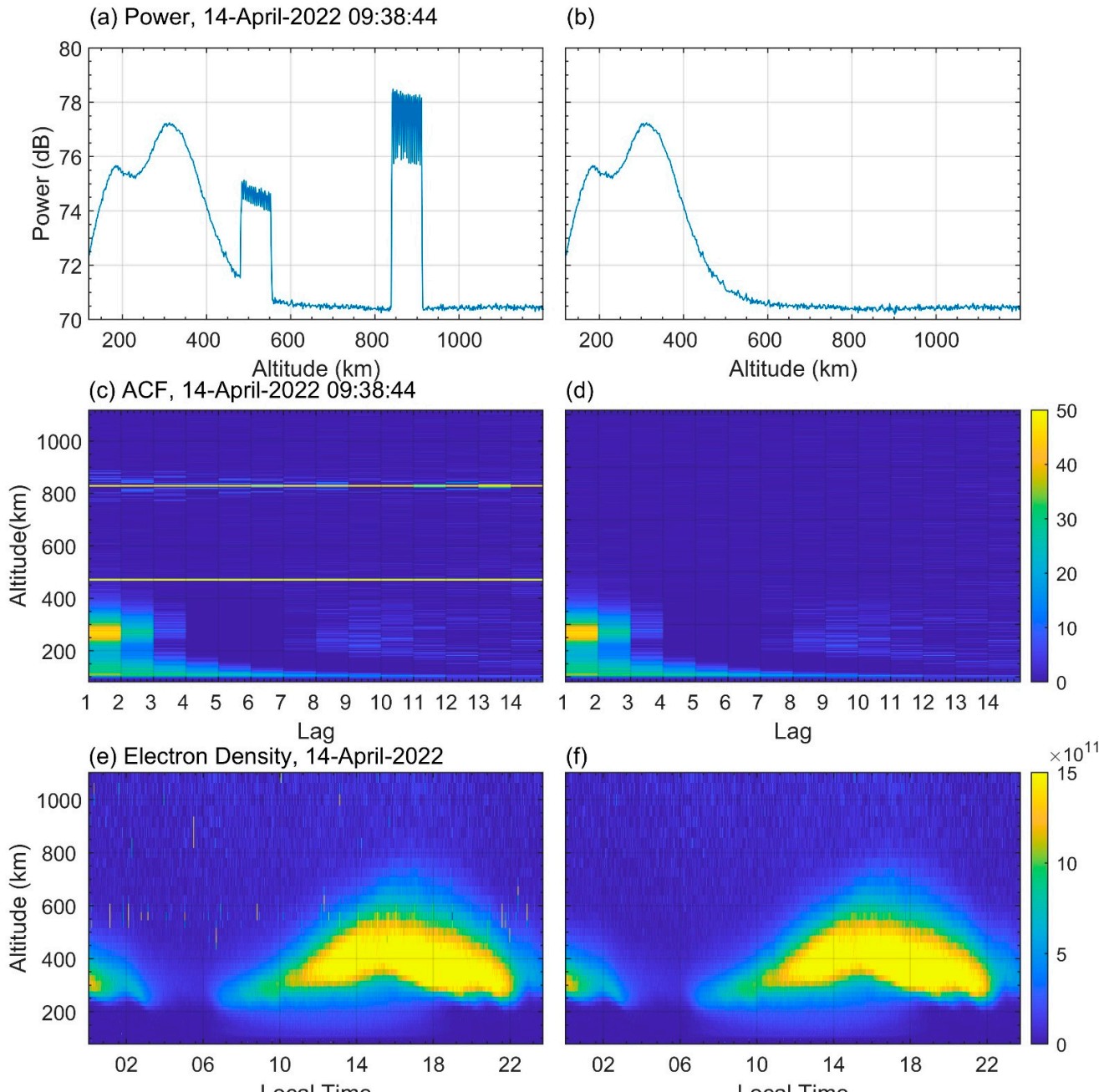

**Figure 4.** An example of power profile ((**a**,**b**) unit: dB) and autocorrelation function ((**c**,**d**) ACF) of SYISR ionospheric measurement observed on 14 April 2022 at 09:38:44 using an alternating code (AC) signal. Panels (**e**,**f**) show the retrieved electron density profiles during the entire day of 14 April 2022. The left (right) column shows the results before (after) eliminating the space object pollution.

This pollution elimination algorithm is suitable for obvious space objects. However, it has limitations, even when integration is employed. We use the difference level between space object echoes and the background signal to determine the space object pollution. When the difference in power is near the fluctuation level of the ionosphere itself, it is difficult to distinguish them. At an altitude with a strong ionospheric background echo, such as the F2 peak region, the detection of space objects with low power is easily missed. Figure 5 shows an example of this case. The space object with low power appears at the altitude region of the ionosphere peak. The increase in power is approximately 1 dB. In the ACF and power spectrum profiles, its effect is weak and difficult to determine. After the

inversion, the space object pollution rises to an abnormal increase in peak electron density in the results. This type of space object pollution remains in the observations and results in unrealistic ionospheric results.

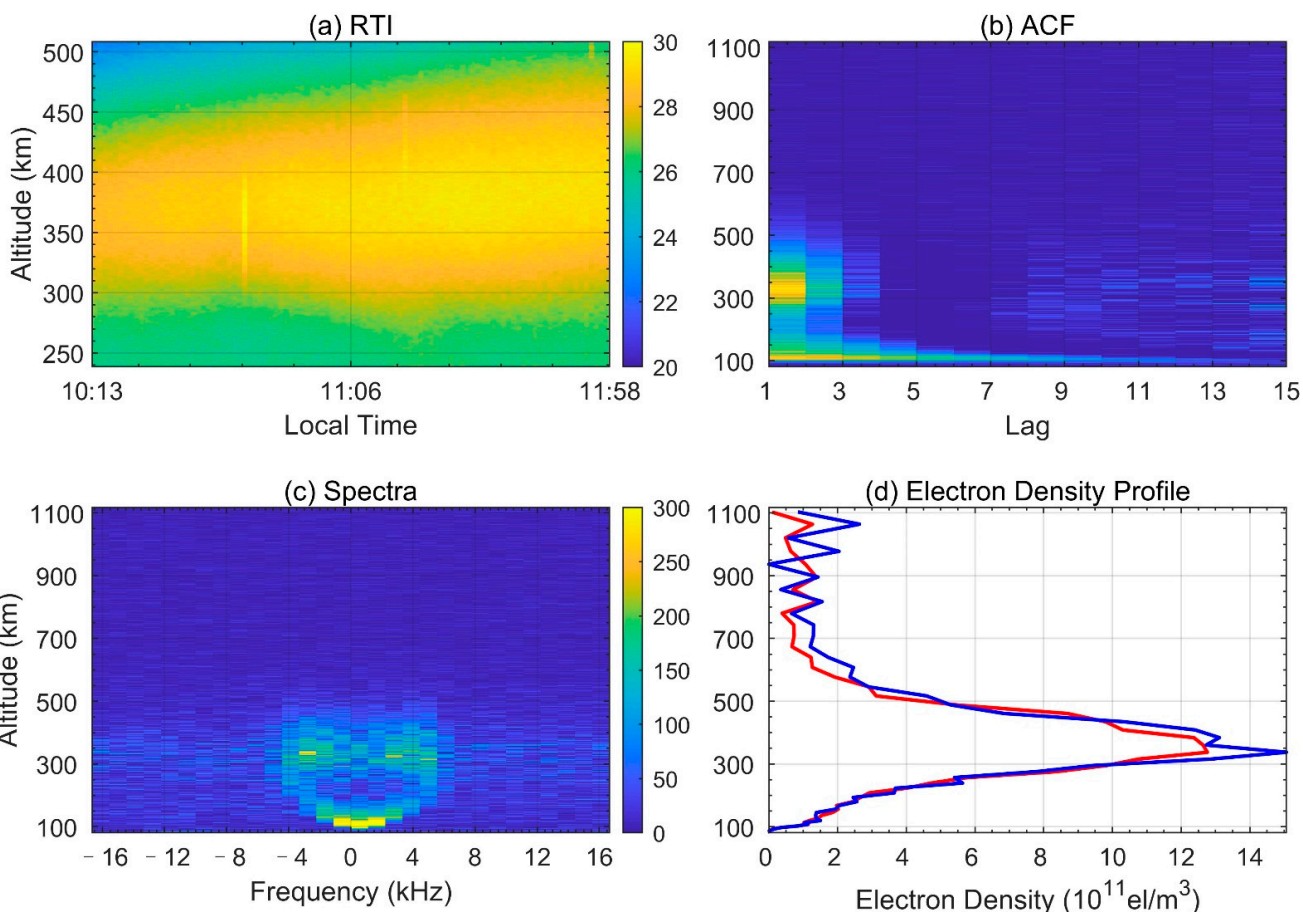

**Figure 5.** (**a**) Power profile (unit: dB) with weak space object pollution in the ionosphere observed at 10:43:53 on 16 April 2022. (**b**) Autocorrelation function (ACF) with the space object pollution. (**c**) Power spectrum with the space object pollution. (**d**) Electron density profile from the measurement with the space object pollution (blue). The red line is the adjacent electron density profile without space object pollution, which is shown as a reference.

## 4. Assessment of the Effect of Constellation Satellites on SYISR Observations

A larger constellation is supposed to cause big problems for the ISR ionospheric observations. To quantitatively assess the effect of the constellation satellite number on SYISR observations, we use SpaceX's StarLink as an example here. StarLink is a large satellite constellation that provides high-speed internet communication services. The first generation of StarLink consists of approximately 12,000 satellites. It started launching in mid-2019 and will finish in 2024. The second generation of StarLink is set to be accomplished in 2027, and the number of satellites will increase to 30,000 [6]. Thus, this constellation plan will add more than 40,000 satellites to orbit. It now consists of a total of 2481 satellites in orbit. The maximum and minimum orbit inclinations are 97.5° and 53°, respectively. We used the latest two-line element (TLE) information of StarLink [28] to perform the assessment. We only considered the effective detection area of SYISR (18.3°N, 109.6°E), whose latitude is free from the orbit inclination limit of the StarLink satellites. The assessment results are shown in Figure 6. The left panel shows the number of StarLink satellites that pass through the SYISR detection region during Days 1–10, which increases from ~9000 to ~90,000, with 90% occurring at altitudes above 500 km. Thus, the pollution on SYISR measurements will be 10 times worse if the satellites increase by 10 times, especially at higher altitudes. We

suggested the data were more easily disturbed above 500 km because the ionosphere echo was usually weaker above this altitude. The right panel shows the number of StarLink satellites that passed through the SYISR detection region over 10 days in 2019–2022. This number increased from ~2000 to ~9000, which implies dramatically increased pollution on SYISR measurements due to the increased number of satellites in orbit each year. As a reference, the launched StarLink satellites during each year are 55, 734, and 957 in 2019, 2020, and 2021, respectively [28].

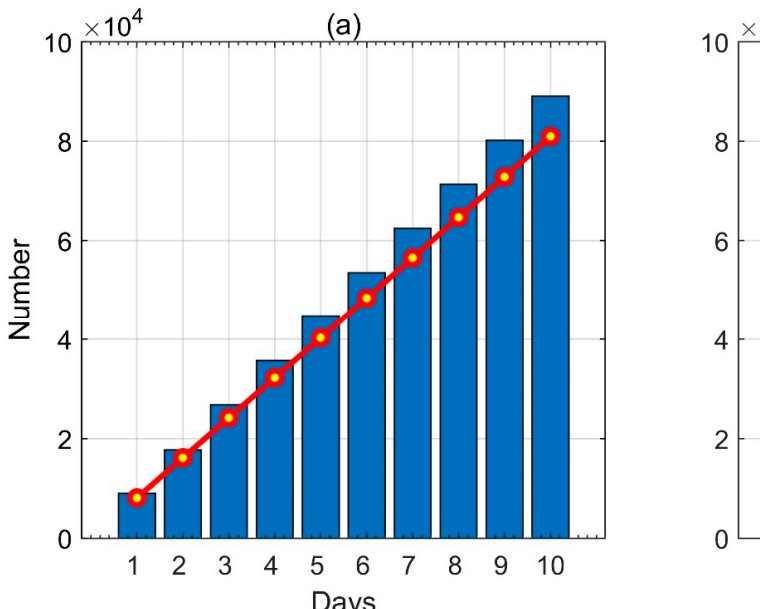
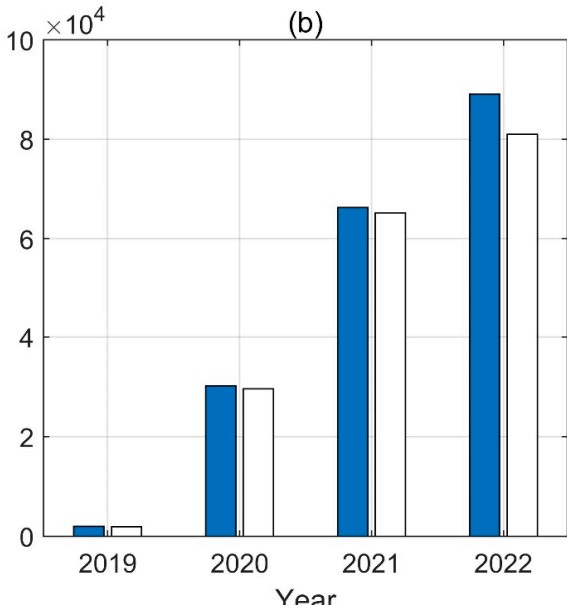

**Figure 6.** (**a**) Number of satellites that pass through the SYISR with different accumulated days based on the latest StarLink constellation orbits. The red line is the number of satellites that appear at altitudes above 500 km. (**b**) The yearly variations of satellite numbers that pass through the detection region of SYISR over 10 days (the blue bar is the total number, and the white bar is the number at altitudes above 500 km). The orbit information is based on the latest StarLink constellation satellites.

When a space object passes through the SYISR's beam, the power of the reflected echo is decided by its radar cross section (RCS), which is related to the size of the object, and the range between the object and the radar receiver. As shown in Figure 5, if the space object is too small to be eliminated by the SO-CFAR algorithm, its pollution will remain in the ionospheric observations and result in unrealistic ionospheric parameters. We performed a simulation based on the radar equation to quantify this "dangerous" situation, as shown in Figure 7. The altitude range of the radar measurements is from 100 km to 1200 km. Considering that the ionosphere echo is a kind of random signal, the incoherent integration and special waveform, which make a balance of range resolution and signal power, are used in the ionosphere measurements. The RCS limitation of the space objects described there is calculated using incoherent integration and the according ionosphere detection signal parameters. Moreover, we set the false alarm possibility to avoid confusing the ionosphere echo with the space objects. An RCS smaller than 0.08 $m^2$ is difficult to remove with the integration of 320 pulses (~5 s) at altitudes above 600 km. According to the data published by CeleStrack [28], approximately 7465 cataloged space objects, whose RCS are known in orbit, satisfy the condition. This implies that ~31.47% of the total (~23,720) still affect the ionospheric inversions after the SO-CFAR algorithm. At altitudes below 600 km, the number of space objects with pollution risk is approximately 1336 (5.63% in a total of 23,720).

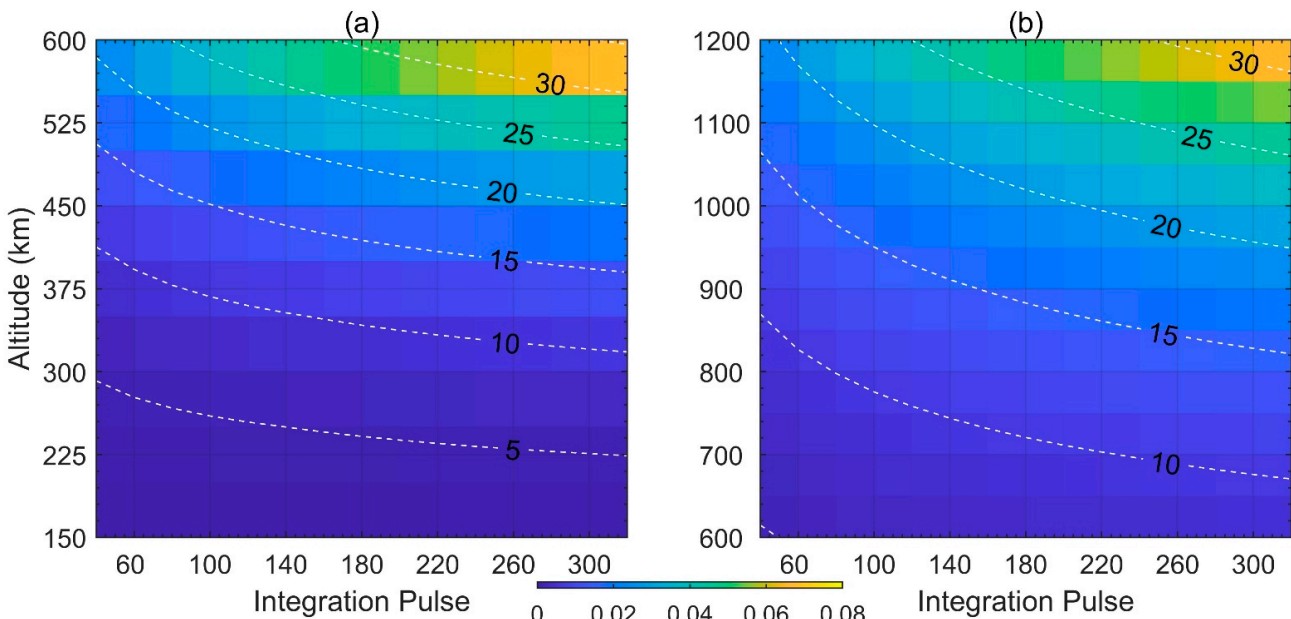

**Figure 7.** Simulated radar cross section (RCS, color map) and the equivalent spherical diameter (ESD, dotted line) of the space objects that retain the risk of polluting the ionosphere observations by SYISR versus integration pulse number and altitude based on the radar equation. (**a**) The range is 150–600 km with a high ionospheric background (SNR = 6 dB), and the detection threshold of the pollution elimination algorithm is four times that of the background ionosphere. (**b**) The range is 600–1200 km with a low ionospheric background (SNR = 0 dB), and the detection threshold of the pollution elimination algorithm is two times that of the background ionosphere. We used the pulse width of 480 μs in the simulations.

## 5. Conclusions

In this study, we drew attention to the effect of greatly increased existing and potential (planned-to-be-launched) LEO constellation satellites on ground-based radar ionospheric observations, where we took the newly built SYISR as an example. We first showed the effect of space objects in the RTI, ACF/power spectra profiles, and ionospheric parameters. Subsequently, we attempted to eliminate space object pollution using the SO-CFAR algorithm. However, pollution from smaller space objects remained, whose reflected echo is comparable to or lower than the background ionosphere, which results in inaccurate findings for the retrieved ionospheric electron density. We quantitatively assessed the space object effect based on the current space object orbit database using a simulation. There remain ~9000 (~37% of the total number of space objects) space objects, whose effect is difficult to eliminate. This pollution should linearly increase when the space object number increases in the future. This study clarifies the effect of space objects on ionosphere observations by the SYISR and is beneficial both to our signal process and to ionospheric parameter inversion, given the frequent occurrence of space objects around the SYISR.

**Author Contributions:** Conceptualization, J.W. and X.Y.; Data curation, J.W.; Formal analysis, J.W.; Funding acquisition, X.Y.; Methodology, J.W.; Project administration, X.Y.; Supervision, X.Y. and F.D.; Visualization, J.W. and X.Y.; Writing—original draft, J.W.; Writing—review & editing, X.Y., F.D., B.N., L.J., C.K., N.Z., J.L., Y.W., H.Y., M.L. and Y.C. All authors have read and agreed to the published version of the manuscript.

**Funding:** This research was funded by the Project of Stable Support for Youth Team in Basic Research Field, CAS, grant number "YSBR-018", the B-type Strategic Priority Program of CAS, grant number "XDB41000000", the National Natural Science Foundation of China, grant number "41427901" and Meridian Project.

**Data Availability Statement:** The experiment results in the figures are available at https://osf.io/2m3tf/. The Two-Line Element (TLE) database and RCS information of space objects in this paper is published at the CeleStrack website (https://celestrak.com).

**Conflicts of Interest:** The authors declare no conflict of interest.

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
