# Peer review of "The Effect of Space Objects on Ionospheric Observations: Perspective of SYISR"

_remotesensing, doi:10.3390/rs14205092_

Round 1
Author Response
Response to Reviewer 1 Comments:
This manuscript reports the echoes from space objected received by the Sanya incoherent scatter radar, which severely affect the data process and retrieval of the ionospheric profiles. The authors attempted to use the SO-CFAR algorithm to remove the contaminated data and assessed the increasing interferences from fast expansion of the low orbit satellites constellations. The method is practical and the results are satisfactory, so the manuscript should be considered for publication. However, I have several minor comments for the author.
Thank you very much for your recognition of our manuscript. We really appreciate your detailed comments. We have completed a minor revision in our manuscript accordingly. The details are presented below:
Point 1: What is the meaning of the red and blue lines in Figure 1(c, f, i)?
Thank you for your comment. The lines in different color represented the power profiles with different incoherent integration time. The red was the results with 32-s integration and the blue was the results with 3.2-s integration. We added a description in the manuscript (line 121).
Point 2: Why does the author show accumulated number of satellites during 1 to 10 days? It seems to be just a simple linear relation. Figure 6 shows that 90% of the passing of StarLink is above 500 km, do the author intend to say that the satellites above 500 km or below 500 km have more influence on the data quality?
Thanks much for the comment. The purpose to do the simulation during 1 to 10 days is to approximate the increase of satellite number on the same orbit. Yes, we suggest the data quality is easier to be disturbed above 500 km. Because the ionosphere echo is weaker at the altitude above this altitude. We added an explain about this in the manuscript (line 254-256).
Point 3: The results indicate that space objects with RCS smaller than 0.08 square meters are difficult to remove from the data, however, there should be a lower limitation that the objects with RCS smaller than that can not be observed by the radar. What is the lower limitation? The statistics of the objects that should exclude the objects with RCS below the lower limitation.
Thank you for the comment and the advice. The ionosphere echo is a kind of random signal. The incoherent integration and special waveform that make a balance of rang resolution and signal power was used for the ionosphere measurements. The RCS limitation of the space objects described there was calculated by the way of incoherent integration and the according ionosphere detection signal parameters. And we set the false alarm possibility to avoid regarding the ionosphere echo as the space objects. This caused the higher low detection limitation. We added an explain about this in the manuscript (line 276-281). And for the traditional detection of space objects by SYISR with coherent integration and chirp signal, the theoretical low RCS limitation for the space object detection using 2ms signal at 600 km was about -60 dBsm with a 200 pulses integration. More detail discussions were presented in another paper called “Simulation and Observational Evaluation of Space Debris Detection by Sanya Incoherent Scatter Radar” (https://agupubs.onlinelibrary.wiley.com/doi/10.1029/2022RS007472), which was also cited in the reference 14.
Point 4: In the final part of the analysis, the author gets approximately 7465 cataloged space objects with RCS smaller than 0.08 square meters. How many of the objects are in the altitude range of the radar signal, like from 150 to 1000 km? Could the author provide an altitude distribution of this kind of objects, which will be helpful for the quality evaluation of the future SYISR data.
Thank you much for the comment and the advice. The altitude range to obtain the number of the space objects with the RCS smaller than 0.08 square meters is consistent with the altitude range in the simulation, which is from 100 to 1200km. We added a description about the altitude range in the line 275 of the manuscript. We did not provide the altitude distribution of these objects, because they are not our measured results. They are open published orbital data in the CeleStrack website (https://celestrak.org/#satellitedata/catalog).

Reviewer 2 Report
The problem with disturbances of space objects in incoherent scatter data is well-known in the community and actions is normally taken to reduce this. The techniques used at various radar facilities are not widely published, and are often an integral part of the analysis process. To highlight the problem, as this paper does, is since long awaited. However, here the authors are only skimming the subject, and the main result of the paper is showing the problem, but not really solving it. First is the the detection algorithm. A SO-CFAR algorithm is used, but it's not described how it's actually applied to the data. Examples of false alarms would for example be a great value. I assume this will also depend on aspect angle as the ionosphere is more-or-less ordered by the magnetic field, and by crossing the beam over field lines abrupt natural changes are more common. Almost all debris are following circular orbits and when doing measurements off zenith the changes in range will not be as sharp as the object moves between radar pulses. Secondly, it's the repairing algorithm. The authors are replacing the echo area with data from the neighbourhood. They don¨t explain if this is in time or in space. A better way, which is used at some other radars is just to omit the polluted data for that specific time interval, a few seconds, and take this into account in the following analysis with a reduced number of estimates for the range in question. I would also like to see a discussion what the effect is on the spectral parameters and how that is spread in range with the length of the radar pulse. This problem is not only for LP, but also for AC as an echo disturbs both the decoding process and contributes to the variance over a larger range. In a paper like this, I would like to see at least couple examples of detection algorithms and their benefits, and a wider discussion how to handle the problem.
Author Response
Response to Reviewer 2 Comments:
The problem with disturbances of space objects in incoherent scatter data is well-known in the community and actions is normally taken to reduce this. The techniques used at various radar facilities are not widely published, and are often an integral part of the analysis process. To highlight the problem, as this paper does, is since long awaited. However, here the authors are only skimming the subject, and the main result of the paper is showing the problem, but not really solving it. First is the detection algorithm. A SO-CFAR algorithm is used, but it's not described how it's actually applied to the data. Examples of false alarms would for example be a great value. I assume this will also depend on aspect angle as the ionosphere is more-or-less ordered by the magnetic field, and by crossing the beam over field lines abrupt natural changes are more common. Almost all debris are following circular orbits and when doing measurements off zenith the changes in range will not be as sharp as the object moves between radar pulses. Secondly, it's the repairing algorithm. The authors are replacing the echo area with data from the neighborhood. They don’t explain if this is in time or in space. A better way, which is used at some other radars is just to omit the polluted data for that specific time interval, a few seconds, and take this into account in the following analysis with a reduced number of estimates for the range in question. I would also like to see a discussion what the effect is on the spectral parameters and how that is spread in range with the length of the radar pulse. This problem is not only for LP, but also for AC as an echo disturbs both the decoding process and contributes to the variance over a larger range. In a paper like this, I would like to see at least couple examples of detection algorithms and their benefits, and a wider discussion how to handle the problem.
Thank you very much for the comments and the advice about our work. They are really valuable for us. We revised the manuscript accordingly. The details are presented below:
Point 1: First is the detection algorithm. A SO-CFAR algorithm is used, but it's not described how it's actually applied to the data. Examples of false alarms would for example be a great value. I assume this will also depend on aspect angle as the ionosphere is more-or-less ordered by the magnetic field, and by crossing the beam over field lines abrupt natural changes are more common. Almost all debris are following circular orbits and when doing measurements off zenith the changes in range will not be as sharp as the object moves between radar pulses.
Thank you much for the comment, which reminds us that our description about the method is not insufficient. The detection of the space objects was finished in the step when we obtained the power profiles of the pre-integration pulses, which belongs to a part of the SYISR data process to obtain the ionosphere parameters. If the space objects were detected in some range, the raw IQ data at the according range would be replaced. Considering to the ionospheric fluctuation along the altitude, we divided the background to detect the space objects into a high background and a low background by the altitude of 600km. According to the detection threshold and detection reference unit number used, the theory false alarm probability was about 2% in the high background and 10% in the low background. And because the number of reference unit was set to change with the sampling points of one pulse width, there was a little difference using different waveform. As for the measurement from different azimuth and elevation, the range was change to the altitude for the divide of background. The method was applied in the same way as the radar was pointing to the zenith angle. We added this part in the manuscript for a clearer description for the process (line 151-155, 189-195). As for the SO-CFAR algorithm application in the detection of the space objects, we presented the process in the session 4 and analyzed the detection threshold and reference unit number set in the algorithm (line 150-187).
Point 2: Secondly, it's the repairing algorithm. The authors are replacing the echo area with data from the neighborhood. They don’t explain if this is in time or in space. A better way, which is used at some other radars is just to omit the polluted data for that specific time interval, a few seconds, and take this into account in the following analysis with a reduced number of estimates for the range in question.
Thank you for the comment. We mentioned that the neighboring data at the same range is used to fill the lost data in the manuscript (line 197-199). This was not clear enough and easy to cause the misunderstanding. We added an explain that the filled data is from the time domain (line 198). And the reason we choose to fill the lost data but not just omit this part is for the integrity of the results.
Point 3: I would also like to see a discussion what the effect is on the spectral parameters and how that is spread in range with the length of the radar pulse. This problem is not only for LP, but also for AC as an echo disturbs both the decoding process and contributes to the variance over a larger range.
Thank you for the detailed comments. The decoded spectrum and ACF results with the space object effect were presented in figure 2. We discussed the different characteristics of power profiles and spectrum using LP and AC in the session 2 (line 127-138). The effect of radar pulse length、baud bits on the decoded results was included. For the fitted ionosphere parameters before and after the removement of the space objects, we shown the electron density measured by a 16-bits (480us) AC code as an example and present the comparison in the last subplot of figure 4. As you said, the disturbance of space objects had a range extension. The width is usually equal to one pulse length before decoding. And as for the AC signal, the range extension would become equal to a baud length after decoding. Yes, although the decoding of AC improved the range resolution to a narrower width for the ionosphere echo, the disturbances of space objects were still found over a range extension. The reason was the power of the space objects would change with time. The range extension of space objects was left over even the decoding process was done. We added an explain for the results in the manuscript (line 205-212).
Point 4: In a paper like this, I would like to see at least couple examples of detection algorithms and their benefits, and a wider discussion how to handle the problem.
Thank you much for the comment. The method presented in the manuscript was tested with a fine result to detect the space objects from the ionosphere measurements in SYISR, while it had the limitation for the small objects. We had tried some other methods such as CA-CAFR, median filter. But the results were not good as the method we presented in the manuscript. Considering our main intention of this paper was to show the increasing effect of the space object pollution on the ionosphere measurements by SYISR and draw the attention about this problem, we did not expand the introduction on the different methods.

Reviewer 3 Report
Review report of the manuscript “Space objects on ionospheric observations: Perspective of SYISR” by Junyi Wang, Xinan Yue, Feng Ding, Baiqi Ning, Lin Jin, Changhai Ke, Ning Zhang, Junhao Luo, Yonghui Wang, Hanlin Yin, Mingyuan Li, Yihui Cai.
General comments:
In this manuscript, the authors presented a comprehensive study on impacts of space objects on ionospheric measurements, which is very interesting. The study mainly focused on how space objects contaminate the measured ionospheric densities. There will be a drastic increase in space objects, particularly in Low Earth Orbit (LEO), therefore the present study is a good step to quantify the effects of such objects on ionospheric measurements.
The paper is well written with some interesting case studies. I have minor comments on the manuscript. Authors may consider revising the manuscript accordingly.
1. Authors have not mentioned any previous attempts to eliminate the errors due to space objects on ionospheric measurements. Are there any previous studies using ISR available? If so, the authors need to mention those studies in the introduction.
2. Authors have given the percentage of space objects pollution, which is very high in number at 600 km altitude. How much will be the expected contamination on the observed ionospheric electron density if uncorrected? A figure or table with a percentage of electron density difference (with and without correction) would be more interesting for readers.
3. Authors mentioned that the echoes due to small space junks cannot be eliminated. Is it possible to quantify this effect on overall measured densities?

Author Response
Response to Reviewer 3 Comments:
General comments:
In this manuscript, the authors presented a comprehensive study on impacts of space objects on ionospheric measurements, which is very interesting. The study mainly focused on how space objects contaminate the measured ionospheric densities. There will be a drastic increase in space objects, particularly in Low Earth Orbit (LEO), therefore the present study is a good step to quantify the effects of such objects on ionospheric measurements.
The paper is well written with some interesting case studies. I have minor comments on the manuscript. Authors may consider revising the manuscript accordingly.
Thank you very much for the positive comments of our manuscript. We revised the paper according to your comments. The details are presented below:
Point 1: Authors have not mentioned any previous attempts to eliminate the errors due to space objects on ionospheric measurements. Are there any previous studies using ISR available? If so, the authors need to mention those studies in the introduction.
Thank you for the comment. The disturbance from the space objects in ISR’s ionosphere measurements was well-known and reported before, which is mentioned in the reference 10 and 11. But the detail analyses about the effect and the process for the space object pollution is sparsely published. We added a previous study about the space objects detection by the power spectrum of ISR measurements for the supplement of background introduce in the manuscript (line 51-52).
Point 2: Authors have given the percentage of space objects pollution, which is very high in number at 600 km altitude. How much will be the expected contamination on the observed ionospheric electron density if uncorrected? A figure or table with a percentage of electron density difference (with and without correction) would be more interesting for readers.
Thank you much for the advice. A figure about the difference of the electron density before and after removement of the detected space objects was added as a subplot in Figure 3 (line 170). Hope it’s helpful to present the effect of space objects on the measured ionosphere parameter clearer.
Point 3: Authors mentioned that the echoes due to small space junks cannot be eliminated. Is it possible to quantify this effect on overall measured densities?
Thank you for the comment. We analyzed the RCS of the “dangerous” space junks at different altitude in the session 4 (line 275-289) to assess the effect from them. But the ionosphere echo is random, it’s difficult to assess the theoretical power of the echo when these space junks occur in the ionosphere measurements. Their effect on the final measured densities is hardly to know. This is also a reason why we suggest that the attention should be paid on the increasing number of the space junks.

Round 2
Reviewer 2 Report
Well, I'm still not so happy about the 'repairing' method. It's much better to just omit the corrupted data. An alternative can be to extract and subtract the hard target echo from the IQ values. The method in the paper in fact means creating artificial data and claiming it to be real.
The high false alarm rates worries me. Have you investigated what is causing them, or have you tried lower the detection thresholds? What confidence levels have been tried?
Author Response
Response to Reviewer 2 Comments:
Well, I'm still not so happy about the 'repairing' method. It's much better to just omit the corrupted data. An alternative can be to extract and subtract the hard target echo from the IQ values. The method in the paper in fact means creating artificial data and claiming it to be real.
The high false alarm rates worries me. Have you investigated what is causing them, or have you tried lower the detection thresholds? What confidence levels have been tried?
Thank you very much for your review and detailed comments of our work. We have completed a minor revision in our manuscript accordingly. The details are presented below:
Point 1: Well, I'm still not so happy about the 'repairing' method. It's much better to just omit the corrupted data. An alternative can be to extract and subtract the hard target echo from the IQ values. The method in the paper in fact means creating artificial data and claiming it to be real.
Thank you much for the comments. Omitting the corrupted data is of course a method to deal with the space objects. Our description may be not clear enough. While the incoherent integration was used for the ionosphere measurements by ISR, the results of an integration unit was polluted if one pulse was disturbed. Meanwhile, the signals were coded with different modulation symbol in the AC code. The decoding of AC needed the sum of a circle of these pulses. The pulse number of one circle was double of the baud bits. Once a space object occurred, the pulses belonged to a circle could not finish the decoding. This situation would become worse when the space objects became more and more. We wanted to present another possibility and tried to fill in the polluted data. We used the data at the same range from the neighboring pulse in the time domain to replace the polluted data by the space objects. The filled data was measured within the pre-integrated time (usually a few seconds). Considering the incoherent integration of ionosphere echo was applied under the assumption of a stable ionosphere during the integration time, we regarded the data from the neighboring pulse within this time as the repeated measurements of the stable ionosphere. Then the data was used to be the substitute of ionosphere echo polluted by the space objects. We added this description for a supplement about the repairing method in the manuscript (line 207-212).
Point 2: The high false alarm rates worries me. Have you investigated what is causing them, or have you tried lower the detection thresholds? What confidence levels have been tried?
Thank you for the comments. The false alarm rates reported meant the possibility that the false detection of space object happened. They depended on the detection thresholds and the number of reference unit. The low detection thresholds would make the false alarm rates increase. Different false alarm rates were tried to find a balance between protecting the ionosphere echo and picking out the space objects. We tried double value of the false alarm rates that we reported, which meant the lower detection thresholds were set. At this situation, if the ionosphere fluctuated greatly, the ionosphere echo was easy to be detected as the space objects falsely. And when we lowered the false alarm rates to half, the detection threshold improved. The false dismissal detection of space object increased, especially when the space objects occurred at the altitude of F layer. Because the space object was mixed with the strong ionosphere echo, the difference between the power of space objects and the background with ionosphere was small. With a high detection threshold, it is easy to cause the miss of detection. We added an explanation about this in the manuscript (line 179-187). The space objects occurred at different altitude, and their RCS were different. We tried to eliminate more space object pollution on the premise of protecting the normal ionosphere echo. Although the method would fail when some small space objects occurred, the obvious space objects were deleted from the ionosphere observations while reserving the normal ionospheric echo (results in Figure 3 as an example). Meanwhile, faced the increasing space object pollution, the related research for the optimization would be continued.
